# Fabrication and Tribological Properties of Mesocarbon Microbead–Cu Friction Composites

**DOI:** 10.3390/ma13020463

**Published:** 2020-01-18

**Authors:** Hai-Xia Guo, Jian-Feng Yang

**Affiliations:** State Key Laboratory for Mechanical Behavior of Materials, Xi’an Jiaotong University, Xi’an 710049, China; guohx2011@163.com

**Keywords:** Cu–matrix composites, mesocarbon microbead, mechanical property, tribological behavior

## Abstract

Graphite–metal composites have been used as friction materials owing to their self-lubricity, which is ascribed to the weak interlayer bonding of graphite. To overcome the shortage of graphite flake (GrF)-filled composites of having low tribological properties, graphite-Cu composites with mesocarbon microbead (MCMB) as the solid lubricant are developed in this paper. The MCMB–Cu composites have a lower friction coefficient and wear rate than do the GrF–Cu composites taken as reference materials, exhibiting a better self-lubricating performance. Microstructural analysis indicates that the relatively weaker interlayer bonding of the MCMB, smooth interface between the MCMB and matrix, and more cementite formation thorough reaction of MCMB and iron are the key factors behind the enhanced tribological properties. In addition, both the friction coefficients and wear rates of the two groups of composites gradually decrease with the graphite content. This work opens an avenue for designing desirable graphite-based metal friction materials.

## 1. Introduction

Copper and its alloys have high potentials for use in tribological applications, such as electrical contacts, bearings, and bushings, owing to their superior corrosion resistance and electrical and thermal conductivity [1,2,3]. However, their poor wear resistance and the failure of liquid lubricants under extreme conditions restrict their potential to some extent. Incorporating solid lubricants (such as graphite, MoS_2_, and h-BN) and ceramic particles (i.e., Al_2_O_3_, FeCr, and SiC) into the copper matrix is an effective approach for achieving high wear resistance and self-lubricity [4,5,6,7,8,9]. Graphite has been widely used as a solid lubricant within copper matrices owing to its good lubricity and easy availability. It exhibits differences in the aspects of type, size, and shape, which govern the lubricating properties of the composites. Accordingly, understanding the influences of these factors and the lubrication mechanisms can help in the design of advanced metal matrix composites with desirable self-lubricating performances.

A great deal of effort has been devoted to exploring the effects of graphite on the wear resistance and self-lubricity of Cu–matrix composites. For example, Rajkumar and coworkers [10] investigated the effects of the amount of graphite on the self-lubricity of copper–TiC–graphite composites. They found that Cu–matrix composites with high graphite contents showed good wear resistance. Zhan et al. [11] demonstrated that the wear resistance of copper–SiC–graphite composites increased as the amount of graphite increased under low loads, while the situation reversed with high graphite contents under high loads. Similar results have also been observed by Wang et al. [12] and Chen et al. [13]. In addition to graphite flakes, other kinds of graphite including expanded graphite and graphene nano-platelets have also been used to improve the wear resistance of Cu-based composites [12,14]. In these studies, the types of graphite used were irregular flakes. Their sharp shapes caused obvious stress concentrations at the contact surfaces between the graphite and matrix and interfacial failure under loading, hindering the enhancement of the wear resistance of the composites to a certain extent. On the other hand, a larger interlayer spacing results in weaker interlayer bonding [13] in the crystalline structure of graphite, which leads to easier delamination between adjacent interlayers under loading. In this situation, thinner graphite sheets are sheared off during shear-friction processes, which results in easy formation of a solid lubricant film at the friction surface, thus reducing the wear rates of composites [12]. As a promising type of artificial graphite, mesocarbon microbead (MCMB) has a regular spheroidal shape with a layered structure after being graphitized, and exhibits high hardness and cycle stability [15,16]. Compared with graphite flakes, MCMBs have a smoother shape and a larger interlayer spacing (from 0.3405 nm to 0.3350 nm) [17]. These features provide the possibility of achieving excellent wear resistance by introducing MCMB as a solid lubricant into metal matrices. However, to the best of our knowledge, there has been no reporting on MCMB–metal friction materials.

In this paper, therefore, MCMB–Cu matrix composites are developed by vacuum hot-pressing technology, and their tribological properties are investigated for the first time, with graphite flake (GrF)–Cu composites used as the reference material. MCMB–metal matrix composites exhibit superior self-lubricating performance in comparison with GrF–Cu composites. This work opens an avenue for the design of graphite-based metal friction materials.

## 2. Experimental

### 2.1. Materials

The spherical atomized copper (mean particle size: 2 μm, Shanghai Chaowei Nanotechnology Co., Ltd., Shanghai, China), elliptical reduced iron (mean particle size: 3 μm, Shanghai Chaowei Nanotechnology Co., Ltd., Shanghai, China), and rod-like aluminum (mean particle size: 57 μm, Shanghai Chaowei Nanotechnology Co., Ltd., Shanghai, China) of 99.9% purity were mixed, and the mixture was used as the matrix of the MCMB–Cu composites. The particle size of alumina (~100 μm magnitude) is frequently used as a reinforcement in metal matrix composites [18]. Therefore, clumpy alumina (mean particle size: 150 μm, Sinopharm Chemical Reagent Co., Ltd., Shanghai, China) of 99.0% purity was employed as a reinforcement to improve the mechanical properties of the composites. Graphite flakes are the most common lubricant used in Cu–matrix composites, and particle sizes of 30–50 μm are widely studied in metal matrix composites [4,6,10]. To reflect the improvement of the tribological properties of the MCMB–Cu composites developed in the present work, graphite flakes (mean particle size: 40 μm, Qingdao Jin Ri Lai Graphite Co., Ltd., Qingdao, China) of 99.0% purity were selected to prepare GrF–Cu composites as reference materials. MCMB (mean particle size: 40 μm, Tianjin Eminent Battery Material Co., Ltd., Tianjin, China) of 99.0% purity was used as the solid lubricant. In this case, the MCMB had a nearly spherical shape, while GrF had a cuboids shapes with approximate dimensions of 40 μm × 10 μm × 1 μm. The micromorphologies of the two types of graphite particles are shown in Figure 1.

### 2.2. Preparation Method

Vacuum hot-pressing technology was applied to prepare the MCMB–Cu composites. This is a method often used in the preparation of metal matrix composites [13,19]. The preparation process was as follows. First, the copper, iron, and aluminum particles were wet-milled for 6 h in a plastic bottle using high-purity alumina balls as the grinding medium and anhydrous alcohol as the dispersing agent. The resulting slurry was vacuum-dried in a rotary evaporator. Then, the dried mixture, MCMB, and alumina were dumped into in a V-type mixer and mixed for 30 min. The powder mixture was poured into a graphite mold with a cylindrical groove, whose diameter and height were 30 mm and 8 mm, respectively. The mold was deposited in a vacuum furnace (Highmulti-5000, Fujidempa Co. Ltd., Osaka, Japan) and sintered at 980 °C for 1 h under a uniaxial pressure of 20 MPa. Then, the power was turned off, and the sample temperature was decreased from 980 °C to 250 °C at a cooling rate of 4 °C/min in the furnace. Finally, the sample was removed from the furnace at room temperature after furnace cooling. The GrF–Cu composites were fabricated by the preparation process used for the MCMB–Cu composites. The chemical compositions of these two composites are listed in Table 1. Depending on the different solid lubricants and their contents, the samples of these composites were named *solid lubricant*-n, in which *solid lubricant* includes MCMB and GrF and n denotes the solid lubricant volume content. For example, MCMB-5 represents the MCMB–Cu composite with 5 vol.% MCMBs.

### 2.3. Characterizations

To characterize the crystalline structures of MCMB and GrF, their diffraction patterns were measured using X-ray diffraction (XRD) by means of an X-ray diffractometer (PW 1730, Philips, Eindhoven, The Netherlands) with monochromatic CuKα radiation at 40 kV and 40 mA. The samples were scanned in the 2*θ* interval of 10°–90° at a scanning speed of 0.3°/s with Si as an internal standard. The Bragg equation is as follows:(1)d = nλ/2sinθ
in which *d* is the interplanar spacing; *θ* is the Bragg angle; the diffraction series *n* = 1, 2, 3…; and the wavelength of the X-rays *λ* = 1.5406 nm. When the diffraction angle (2*θ*) of the diffraction peak corresponding to the MCMB and GrF crystal plane was obtained, the interlayer distance *d* of the MCMB and GrF could be obtained by Equation (1), in which the diffraction series (*n*) took the value of one.

The porosity and bulk density of the sintered products were measured by the Archimedes displacement method with distilled water as the immersion liquid.

The hardness values of the bulk prepared materials were measured using a Brinell testing machine (HBW-3000A, Shanghai Lianer Testing Equipment Co. Ltd., Shanghai China) with an indenter diameter of 5 mm. The applied load and holding time were 250 kg and 30 s, respectively. At least five replicates were performed for the hardness test for each specimen, and average values were obtained.

The flexural strengths of the samples were obtained by cutting, grinding, and polishing to dimensions of 2 mm × 2 mm × 25 mm. The flexural strength was measured on an Instron 1196 machine (Instron Corporation, Boston, MA, USA) by the three-point bending method, with a 20.0 mm span and a cross-head speed of 0.5 mm/min. The specimen tensile side for the strength test was normal to the hot-pressing direction. At least five specimens were tested for each condition, and the standard error was calculated.

Scanning electron microscopy (SEM) (Phenom proX, Phenom-World BV, Eindhoven, The Netherlands) was used to examine the microstructures of the composites after etching in 4% nitric acid–alcohol solution.

The coefficients of friction and the wear characteristics of the Cu matrix composites were evaluated using a ball-on-disk tribometer (LFT-1, Lanzhou Zhongke Kaihua Technology Development Co., Ltd., Lanzhou, China). A commercially available GCr15 bearing ball with a measured average hardness of HRC61 and a diameter of 5 mm was used as the counterpart ball, and the prepared sample was used as the disc. Before testing, the composite disk samples and balls were cleaned using acetone in an ultrasonic bath for 15 min, followed by drying in a vacuum oven at 70 °C for 0.5 h. The bearing ball was kept stationary under an applied load of 50 N, while the composite specimens were fixed on a disk in the tribometer. The disk could be actuated to repeatedly move along the surface with a constant stroke of 5 mm and a frequency of 500/min, which was the equivalent of a linear speed of 80 mm/s. All tests were carried out in air at room temperature (RT) with a humidity of 60%. A sliding time of 30 min was used for each test. The wear tests were carried out in triplicate. The coefficient of friction was determined by an analog-to-digital converter, and simultaneously recorded by a computer during the wear test. The wear rate was calculated by the following equation:*W* = *m*/*ρLN*(2)
where *m* is the mass loss of the sample; *ρ* is the density of the sample; *L* is the total movement distance of the disk during the test; and *N* is the pressure load applied on the bearing ball, with *N* = 50 N.

The wear debris was collected after the friction test was completed. The morphologies of the frictional surfaces (worn surfaces) and wear debris were determined by SEM analysis to study the wear mechanisms.

## 3. Results and Discussion

### 3.1. Microstructure Characterization

Figure 2 reveals the microstructures of the MCMB–Cu composites. Figure 2a shows the SEM image of a typical MCMB-25 sample surface. The MCMB and Al_2_O_3_ particles are uniformly distributed throughout the Cu matrix without aggregation. MCMB or Al_2_O_3_ particles are in tight contact with the matrix, as shown in Figure 2e. There are no obvious defects, such as pores and cracks, at the interfaces between the MCMB or Al_2_O_3_ particle and matrix, indicating sintering compaction of the sample. In addition, Figure 2b shows the SEM image of a typical GrF-25 sample surface. GrF and Al_2_O_3_ particles are also distributed within the Cu matrix. The two kinds of composites exhibit similar structures.

To further prove the compactness of the sample, the densities and porosities of the two groups of composites are shown in Table 2. When the relative density of a composite is as high as 95%, it is a compact material [19]. It can be seen that, when the MCMB or GrF content reached 25%, the relative density of the composite was still as high as 96%. This results showed that the sample was compact. This result also indicated the successful preparation of these composites using the vacuum hot-press method.

The matrix of MCMB–Cu composites is a compound made up of Cu, Fe, and Al particles. During the preparation process, the carbon atoms in the MCMB may diffuse into the matrix and react with the iron phase to form pearlite [20]. The combination of pearlite, Cu, Fe, and Al particles generates a complex matrix structure, which plays a vital role in the mechanical and tribological properties of the composites. To analyze the structure, Figure 2c displays the BSEM image of the MCMB-25 sample. Some pale gray and black laminar particles are distributed uniformly throughout the white matrix. During the BSEM imaging process, if an element in the sample has a higher atomic number, it has a higher brightness in the image. Cu has a higher brightness (i.e., the white region) owing to its larger atomic number than those of the other elements. The lamellar pearlite is composed of the thin lamellas of ferrite and cementite oriented one over the other. This implies that the black/gray lamellar particles are pearlites. As the basic component of pearlites, both cementite and ferrite may be formed by the eutectoid reaction of iron and carbon during the cooling of the samples after sintering, and the carbon content of cementite is far higher than that of ferrite. Therefore, the cementite phase shows a darker color than that of the ferrite phase in the BSEM image. In this manner, the black parts in these lamellar particles are the cementite phase, and pale gray parts are ferrite. The remaining pure black particles in the matrix are aluminum. During the preparation process of the MCMB–Cu and GrF–Fe composites, because the melting point of aluminum (650 °C) is far lower than the sintering temperature of 980 °C, aluminum will form a liquid phase, and penetrate into the inter space of the Cu and/or Fe particles. The Al will increase the density of the composite, and diffuse into the copper to enhance the sinterability of Cu particles.

In contrast, Figure 2d shows the BSEM image of the matrix of the GrF-25 composite. It can be seen that the matrix of the GrF-25 composite has a microstructure similar to that of the matrix of the MCMB-25 composite shown in Figure 2c. Meanwhile, it is found that the iron phase in the matrix of GrF-25 composite mainly exists in the form of a large amount of ferrite (i.e., pale gray region) and a small amount of pearlite (i.e., lamellar particles). Compared with the GrF-25 composite, there are more black/gray laminar particles in the matrix of the MCMB-25 composite. Upon further observation, it turns out that the black parts of these laminar particles in the MCMB-25 composite have larger total areas than those of the GrF-25 composite. This implies that the former contains more cementite than the latter.

To further prove this conclusion, the matrices of the two groups of composites were investigated by energy dispersive spectroscopy (EDS). The EDS analysis results (i.e., chemical distribution maps) are shown in Figure 3c–j. As can be seen from these figures, the matrix carbon content of the MCMB–Cu composite is higher than that of the GrF–Cu composite. These findings suggest that, during the preparation process of the MCMB–Cu composite, more carbon atoms diffuse from the MCMB into the matrix and react with the iron phase, thereby generating more cementite because MCMB might have higher carbon diffusion activity than that of GrF.

To verify this conclusion, an MCMB(GrF)–Fe composite with a volume ratio (MCMBs or GrFs/Fe) of 1:3 was prepared by the preparation process used for the MCMB–Cu composites. The phases present in the composite could be identified by the Vickers microhardness test. Therefore, we measured the Vickers microhardness of the MCMB–Fe and GrF–Fe composites, and the corresponding microstructures are shown in Figure 4. Specially, the Vickers hardness values of the S points, F points, and remaining position are 978 Hv, 87 Hv, and 200–250 Hv, respectively. The diffusion of carbon atoms into the iron generates the formation of cementite, pearlite, and ferrite, and the corresponding standard hardness values are about 800 Hv, 180–250 Hv, and 80 Hv, respectively. Therefore, we concluded that the MCMB–Fe composite mainly consists of pearlite and cementite, while the composition of the GrF–Fe composite mainly includes ferrite and pearlite. Additionally, the chemical compositions of the S and F points were analyzed by EDS. The EDS analysis results are shown in Table 3. It can be seen that the carbon content of the S points is higher than that of the F points. These results confirm that more carbon atoms diffuse from the MCMB into the iron phase and generate more cementite, because MCMBs have a higher carbon diffusion capacity than that of GrF, which is an attribute that will be further discussed below.

The activity of graphite crystals is usually associated with the distance between adjacent carbon atom layers in their crystal structure [17,21]. Here, we evaluate and compare the activities of MCMB and GrF by analyzing their crystal structures using XRD. Figure 5 shows the XRD patterns of the MCMB and GrF powders using silicon as an internal standard. From Figure 5, the angles of the diffraction peaks of Si can be obtained. In fact, Si has standard angles of the diffraction peaks. The exact value of the measurement error was obtained by comparing the measurement angles of the diffraction peak with the angle of the diffraction peak standard. Then, this error was subtracted from the measured angles of the MCMB and GrF powders to obtain the actual angles of the diffraction peaks. It can be seen that the diffraction profile of the (111) reflection of silicon corresponds to the diffraction angle 2*θ* (28.438°). However, the true angle of the (111) diffraction peak is 28.443° [22]. The difference (i.e., measurement error) between the true and measured diffraction angles was obtained. There are some sharp peaks in the diffraction profiles, suggesting that both MCMB and GrF exhibit an ordered graphite crystal structure. In addition, the diffraction angles 2*θ* of the (002) crystal plane corresponding to the MCMB and GrF are 26.468° and 26.514°, respectively. Then, the correct angle of the (002) diffraction peak of the MCMB and GrF was obtained by eliminating the measurement error. Using Equation (1), the interplanar spacings *d*_002_ of MCMB and GrF were calculated as 0.3367 nm and 0.3350 nm, respectively. The interplanar spacing *d*_002_ represents the distance between adjacent carbon atom layers in the graphite crystal structure [22]. Obviously, the interplanar spacing between adjacent carbon atom layers in the MCMB is larger than that in GrF. The crystal structures of the MCMB and GrF are formed by stacked carbon atom layers held together by Van der Waals forces. The number of carbon atom pairs (forming Van der Waals force interactions) per unit area in the interfaces between adjacent layers in the crystal structure of MCMB may be the same as that of the interfaces between adjacent layers in the crystal structure of GrF. The Van der Waals force interactions between adjacent carbon atom layers are affected by the distance between them. Compared with GrF, the crystal structure of MCMB has a greater interplanar spacing, thus leading to a weaker interaction between the carbon atom layers. Therefore, under thermal fields, the carbon atoms in MCMB are more easily moved out of their lattice than are those in GrF, subsequently diffusing into other mediums (i.e., Fe) to form compounds, suggesting that MCMB has a higher carbon diffusion activity than that of GrF. Furthermore, the weaker layer–layer interaction also leads to easier layer separation of the crystal structure of MCMB than that of GrF, thus generating a thinner cut sheet from MCMB when subjected to friction loading.

### 3.2. Mechanical Properties

#### 3.2.1. Brinell Hardness

Figure 6 depicts the Brinell hardness value of the MCMB–Cu composite and GrF–Cu composite at the MCMB and GrF volume contents of 5%, 15%, and 25%. With the same MCMB and GrF contents, the Brinell hardness of the MCMB-filled composites is higher than that of the composites filled with GrF. Specially, the Brinell hardness values of the MCMB-25 composite and GrF-25 composite are 101 and 75 HBW, respectively, and the former is nearly 1.5 times the latter. For the same MCMB and GrF contents, compared with the GrF–Cu composite, more Fe_3_C particles are generated in the MCMB–Cu composite owing to the higher carbon diffusion activity of MCMB than of GrF. The hardness of these Fe_3_C particles is higher than that of other composite components such as Cu, Fe, and Al. Furthermore, the dispersion of Fe_3_C particles in these composites is beneficial in preventing dislocation movement in the crystals of copper and iron and enhancing the deformation resistance during loading [23]. Therefore, a combination of more Fe_3_C particles and their dispersion strengthening effect results in a higher hardness of the MCMB–Cu composites than that of the GrF–Cu composites at the same Al_2_O_3_ content. When the content of MCMB and GrF is 25%, as shown in Figure 2c,d, the total amounts of black parts (i.e., Fe_3_C) in the black/gray laminar particles in the MCMB-25 composite are far higher than those of the GrF-25 composite, resulting in a higher hardness of the MCMB-25 composite. A similar result has also been reported by Ram Prabhu et al. [20], who have shown that, the higher the volume fraction of the reinforcement in the metal matrix, the higher the hardness of the composite. It can also be observed that, as the MCMB content increases from 5% to 25%, the hardness of the MCMB–Cu composite gradually decreases owing to the lower hardness of the MCMB particles [24]. As the MCMB content increases, the copper, iron, and aluminum mixture content decreases to keep the total volume content of all compounds constant. The negative effect of increasing the MCMB content is much stronger than the positive effect of the accompanying Fe_3_C increasing effect on the hardness of the composites. This trend and the associated physical mechanisms are analogous to the effect of the GrF content on the hardness of the GrF–Cu composites. On the basis of Archard’s relation [25], a high hardness value is correlated with a better wear resistance; these trends are of interest while examining the wear rate of the composites.

#### 3.2.2. Flexural Strength

Figure 7 shows the flexural strengths of the MCMB–Cu composite and GrF–Cu composite at the different MCMB and GrF volume contents of 5%, 15%, and 25%. With the same MCMB and GrF content, the flexural strengths of the MCMB-filled composites are higher than those of the composites filled with GrF. When MCMB and GrF particles are present in the composites, the stress concentration factor near the tips of the MCMBs should be considerably lower than that near the flake graphite; thus, under loading, brittle cracking is less intense in the composite containing MCMBs than in that containing flake graphite. Furthermore, dispersion strengthening of the MCMB–Cu composite by Fe_3_C formation is also an important factor in enhancing the MCMB–Cu composite strength. The combined effect of these aspects leads to a higher flexural strength of the MCMB–Cu composites in comparison with that of the GrF–Cu composites. It can also be observed that, as the MCMB content increases from 5% to 25%, the flexural strength of the MCMB–Cu composites gradually decreases. The strength and hardness of MCMB particles are extremely low, so their presence can induce a notch effect in the matrix, which is the likely rupture origin in the composites, and consequently, the flexural strength is reduced at higher MCMB contents.

### 3.3. Tribological Properties

#### 3.3.1. Friction Coefficient

Figure 8 shows the friction coefficients of MCMB–Cu composites and GrF–Cu composites under different MCMB and GrF volume contents. As can be seen in Figure 8, the MCMB–Cu composites have lower friction coefficients than those of the GrF–Cu composites at the same MCMB and GrF content. When the composites are subjected to surface friction loads, plastic deformation of the metallic matrix occurs, and brittle MCMB or GrF around the matrix are extruded to break into small graphite debris, resulting in the formation of a graphite-rich tribofilm on the friction surface of the composites [26]. Chen et al. [13] also proved that the solid lubricant migrates from the matrix to the worn surface and forms a tribofilm mainly containing the solid lubricant during the friction process of metal matrix composites. As the crystal structure of MCMB has a larger interlayer spacing and a weaker interlayer interaction than those of GrF, thinner and smaller MCMB debris are generated under surface friction on the MCMB–Cu composites, as shown in Figure 9. Therefore, a smoother tribofilm is formed on the friction surface, leading to a lower friction coefficient of the MCMB–Cu composites in comparison with the GrF–Cu composites.

The friction coefficient of the MCMB–Cu composites decreases from 0.84 to 0.25 as the MCMB content increases from 5 vol.% to 25 vol.%. As the MCMB content increases, the content of the metal mixture matrix, which provides the main resistance to sliding friction during the friction process, decreases in the composites. This results in a decrease of the friction force and, thereby, the decline of the friction coefficient under the same pressure loading. Meanwhile, the MCMB–Cu composites with a higher MCMB contents generate more MCMB debris, thus forming smoother MCMB-rich tribofilms, which is beneficial to reducing the friction coefficient. The combined effects of these aspects result in a significant reduction of the friction coefficient of the MCMB–Cu composite with increasing MCMB contents.

#### 3.3.2. Wear Rate

Figure 10 shows the wear rates of the MCMB–Cu composites and GrF–Cu composites under different MCMB and GrF volume contents. It can be seen that the MCMB–Cu composites have a lower wear rate than that of the GrF–Cu composites under the same MCMB and GrF content. As mentioned before, compared with the GrF–Cu composites, the metal matrix of the MCMB–Cu composites contains more Fe_3_C particles owing to the higher carbon diffusion activity of MCMB than of GrF. The high hardness of these Fe_3_C particles contributes to a higher hardness of MCMB–Cu composites, showing higher deformation resistance, which results in a smaller compressive deformation of the composites subjected to an initial surface pressure during friction. Meanwhile, these Fe_3_C particles also play an enhancing role in the strength of the MCMB–Cu composites. As the friction process continues, MCMB–Cu composites show higher shear resistance, leading to the formation of some shallower grooves on their friction surfaces, thus reducing the wear rate of MCMB–Cu composites. The high hardness and strength of MCMB–Cu composites play an important role in reducing the wear rate of composites, which has been illustrated in the work of Ram et al. [20]. On the other hand, the shape of the flake GrFs is sharper than that of the spherical MCMBs. Under the same loading, the interfacial stress between the GrF and matrix is higher than that at the MCMB/matrix interface [27], which results in easier crack initiation and propagation [28] and the flaking of more GrFs and metal matrix, forming some holes on the friction surface, as shown in Figure 11a,c. It can be concluded that the flaking in the MCMB-25 composite occurs in smaller areas and at shallower depths compared with that of the GrF-25 composite. Compared with GrF debris, the thinner and smaller MCMB debris can effectively fill the holes of the worn surface of the MCMB-25 composite, which may also be a key factor behind the reduced flaking areas, in addition to the smooth MCMB/matrix interface and more cementite presence. This phenomenon also implies that more uniform and compact tribofilms can be formed on the friction surfaces of MCMB–Cu composites in comparison with GrF–Cu composites. Such a tribofilm facilitates the reduction of the adhesive wear between the MCMB–Cu composites and hard counters. The combined effect of these aspects leads to a lower wear rate of the MCMB–Cu composites in comparison with that of the GrF–Cu composites.

We can also find from Figure 10 that, with the increasing MCMB contents, the wear rates of the MCMB–Cu composites decrease. As has been proven, a higher MCMB content enhances the amount of Fe_3_C generated in the matrix of the MCMB–Cu composites, which is beneficial to reducing their wear rate. A higher MCMB content also leads to the formation of a more uniform and compact tribofilm, weakening the adhesive wear rate of the composites.

At present, GrF–metal matrix friction materials have been widely used in train brake equipment in place of early metal friction materials [29,30]. Their certain self-lubricity addresses the blocking problem that occurs during the braking process using metal friction materials. In this work, we have developed MCMB–Cu composites that might be used as friction materials. Our analyses in Section 3.3 demonstrate that MCMB–Cu composites exhibit lower friction coefficients and wear rates than do GrF–Cu friction materials with the same solid lubricant content, suggesting a better self-lubricity of the former compared with that of the latter. This also implies that MCMB is a promising solid lubricant for use in metal-based friction materials in comparison with GrF. Furthermore, MCMB–Cu composites have higher hardness values than GrF–Cu composites under the same MCMB and GrF content. These results also illustrate that adding a relatively small amount of MCMB allows MCMB–Cu composites to present an equivalent friction coefficient to that of GrF–Cu composites containing more GrF. MCMB–Cu composites also have higher hardness values and strengths than those of GrF–Cu materials. These characteristics are advantageous in designing brake composites with desirable friction coefficients and high hardness and strength through adjusting the content of MCMB.

## 4. Conclusions

MCMB–metal matrix self-lubricity materials were successfully developed for the first time by powder metallurgy. The tribological properties and lubrication mechanisms of the composites have been elucidated. It was demonstrated that composites using MCMB as the solid lubricant have lower friction coefficients and wear rates than do composites containing flake graphite, and both the friction coefficients and wear rates of the two groups of composites gradually decrease with the solid lubricant content. The weaker interlayer bonding of MCMB results in smaller MCMB debris, and consequently, a uniform, compact, and smooth MCMB-rich tribofilm can be formed on the friction surfaces of MCMB–Cu composites, which contributes to the low friction coefficients of the composites. The high activity of MCMB enhances carbon diffusion in Fe to form more Fe_3_C, resulting in the high strength and hardness of the composites and, thereby, the low wear rate of MCMB–Cu composites. As a novel friction material, MCMB–Cu composites could be a promising alternative for use in tribological applications.

## Figures and Tables

**Figure 1 materials-13-00463-f001:**
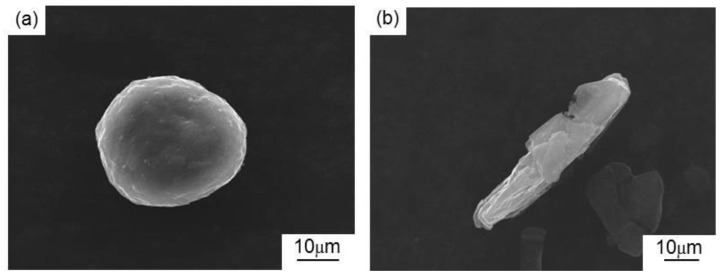
Micromorphologies of the as-received graphite: (**a**) mesocarbon microbead (MCMB) and (**b**) graphite flake (GrF).

**Figure 2 materials-13-00463-f002:**
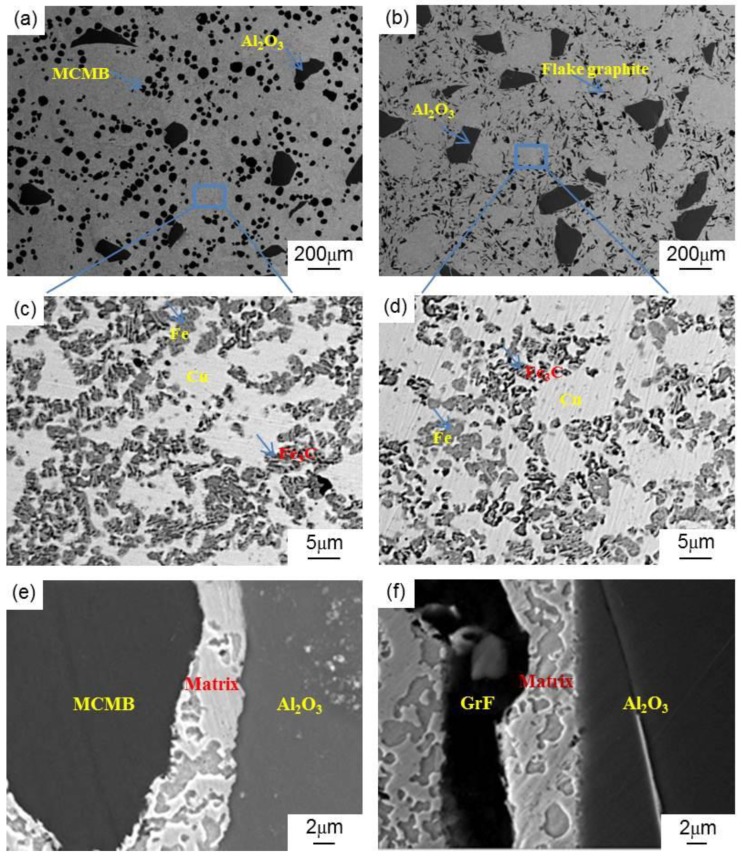
Scanning electron microscopy images (SEM) of the (**a**,**e**) MCMB-25 and (**b**,**f**) GrF-25 composites, and the backscatter electron image (BSEM) of the matrices of the (**c**) MCMB-25 and (**d**) GrF-25 composites.

**Figure 3 materials-13-00463-f003:**
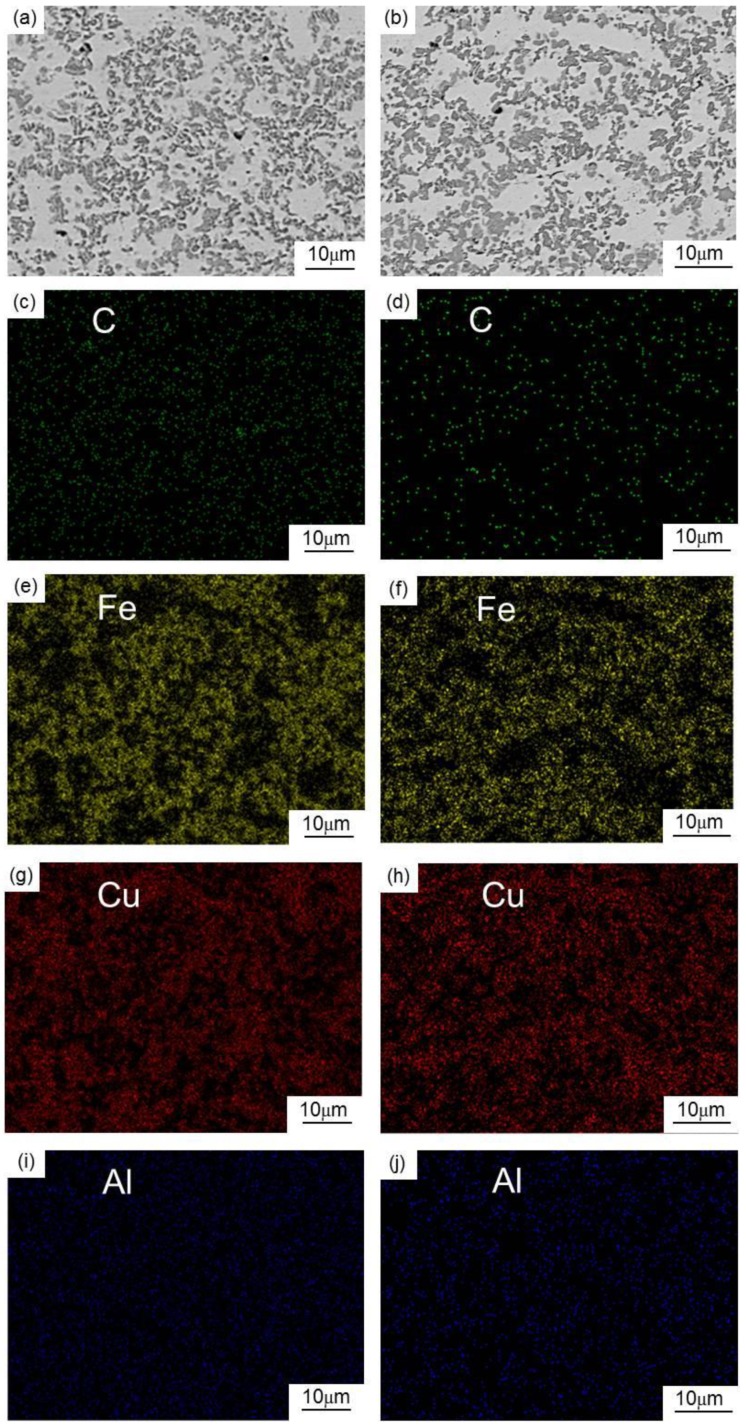
BSEM of the matrices of the (**a**) MCMB-25 and (**b**) GrF-25 composites, and energy dispersive spectroscopy (EDS) maps of the matrices of the (**c**,**e**,**g**,**i**) MCMB-25 and (**d**,**f**,**h**,**j**) GrF-25 composites.

**Figure 4 materials-13-00463-f004:**
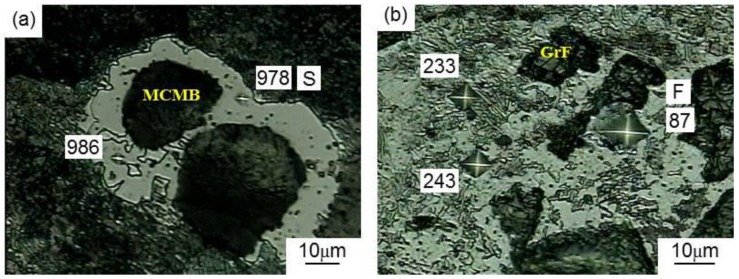
Microstructures of the (**a**) MCMB–Fe composite and the (**b**) GrF–Fe composite.

**Figure 5 materials-13-00463-f005:**
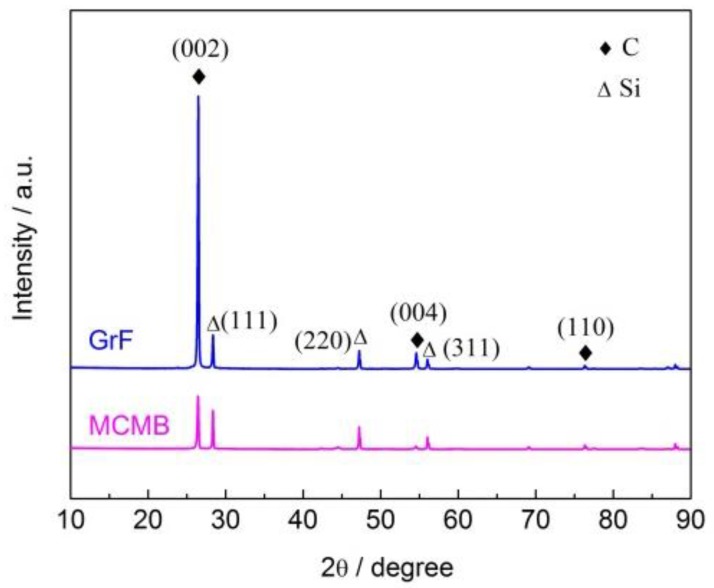
X-ray diffraction patterns of the MCMB and GrF powders.

**Figure 6 materials-13-00463-f006:**
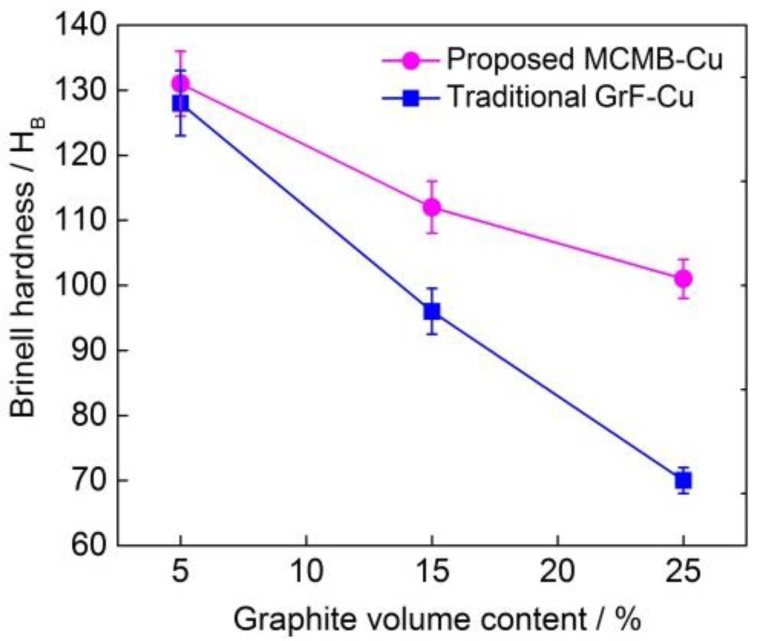
Brinell hardness value of the MCMB–Cu and GrF–Cu composites with different MCMB and GrF volume contents.

**Figure 7 materials-13-00463-f007:**
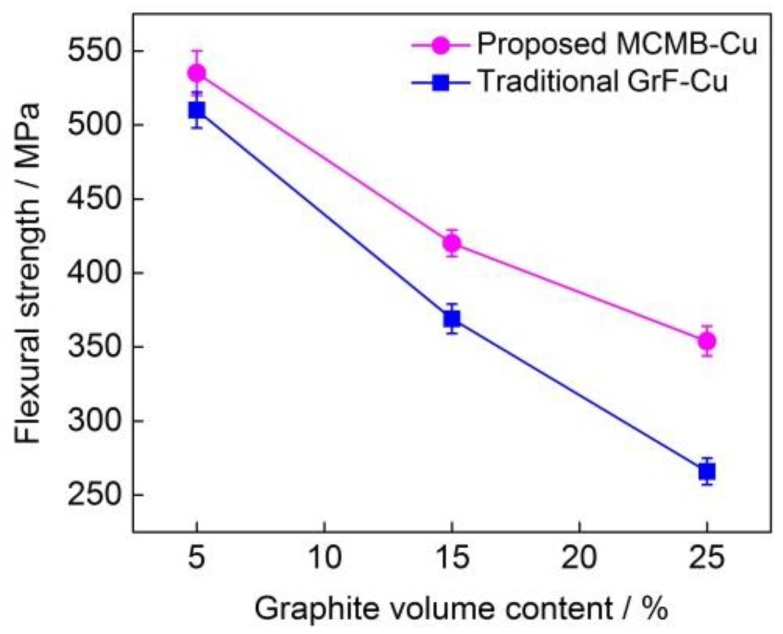
Flexural strengths of the MCMB–Cu and GrF–Cu composites under different MCMB and GrF volume contents.

**Figure 8 materials-13-00463-f008:**
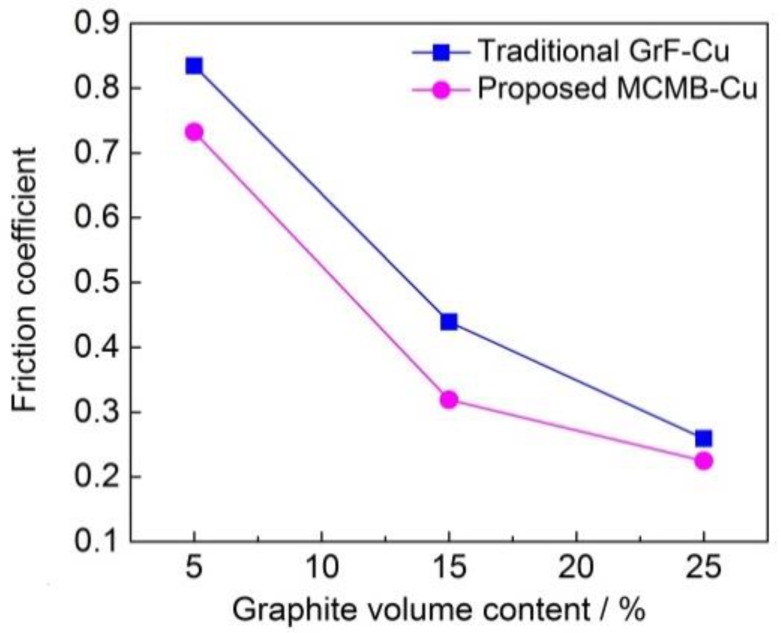
Friction coefficients of the MCMB–Cu and GrF–Cu composites under different MCMB and GrF volume contents.

**Figure 9 materials-13-00463-f009:**
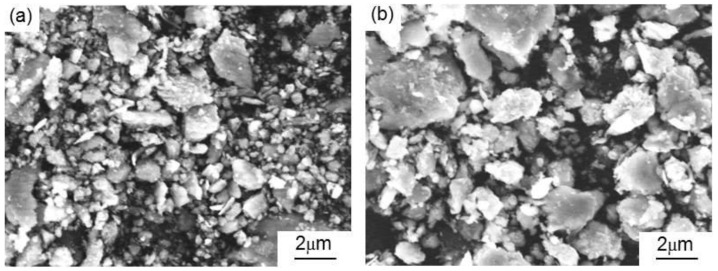
SEM images of the graphite debris of (**a**) the MCMB–Cu composites and (**b**) the GrF–Cu composites.

**Figure 10 materials-13-00463-f010:**
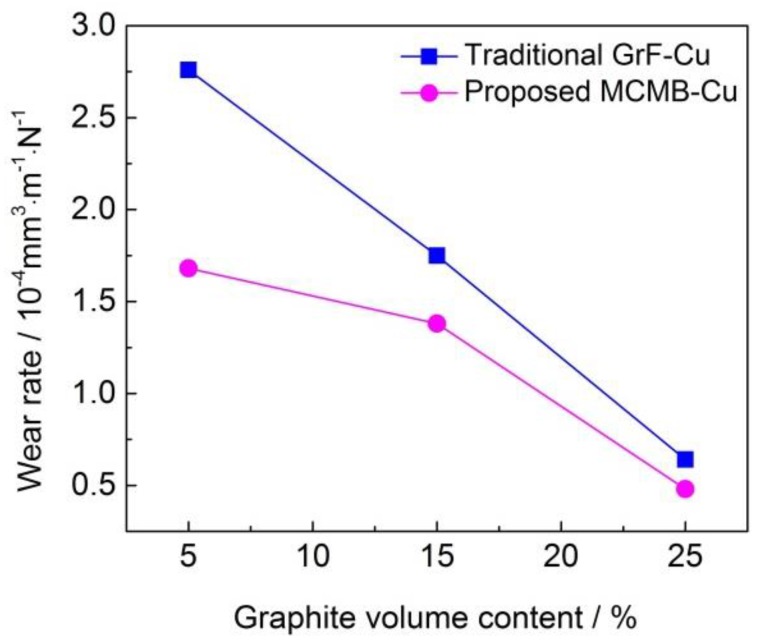
Wear rates of the MCMB–Cu and GrF–Cu composites under different MCMB and GrF volume contents.

**Figure 11 materials-13-00463-f011:**
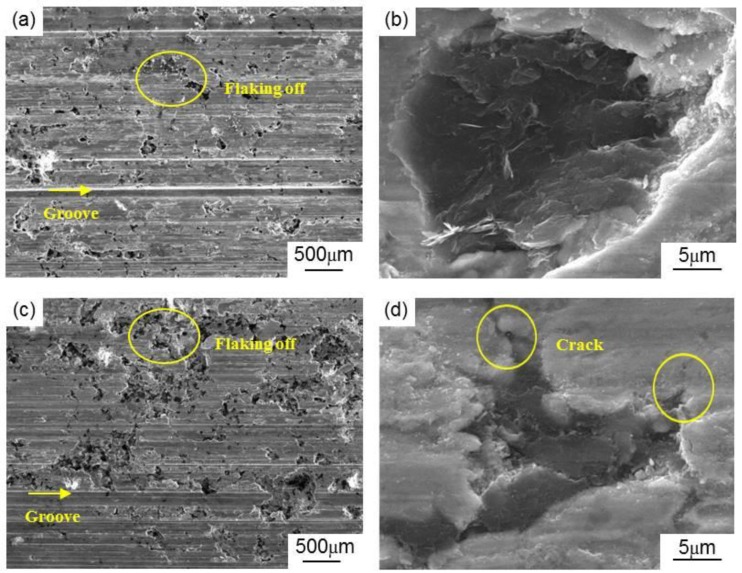
Wear surface morphologies of the (**a**,**b**) MCMB–Cu composites and (**c**,**d**) GrF–Cu composites.

**Table 1 materials-13-00463-t001:** Chemical compositions (vol.%) of the mesocarbon microbead (MCMB)–Cu composites and graphite flake (GrF)–Cu composites.

Samples	MCMB	GrF	Al_2_O_3_	Cu	Fe	Al
MCMB-5	5.0	0	5.0	50.4	34.2	5.4
MCMB-15	15.0	0	5.0	44.8	30.4	4.8
MCMB-25	25.0	0	5.0	39.2	26.6	4.2
GrF-5	0	5.0	5.0	50.4	34.2	5.4
GrF-15	0	15.0	5.0	44.8	30.4	4.8
GrF-25	0	25.0	5.0	39.2	26.6	4.2

**Table 2 materials-13-00463-t002:** Densities and porosities of MCMB–Cu composites and GrF–Cu composites.

Samples	Theoretical Density (g/cm^3^)	Bulk Density (g/cm^3^)	Relative Density (%)	Porosity (%)
MCMB-5	7.62	7.43	97.51	2.49
MCMB-15	7.03	6.95	98.72	1.28
MCMB-25	6.45	6.34	98.29	1.71
GrF-5	7.62	7.50	98.42	1.58
GrF-15	7.03	6.90	98.15	1.85
GrF-25	6.45	6.21	96.27	3.73

**Table 3 materials-13-00463-t003:** Chemical compositions of the different points.

Composition (S Marker)	Element	Weight Percent %	Atom Percent %
	C K	6.59	24.71
	Fe K	93.41	75.26
	total	100	100
**Composition (F Marker)**	C K	0.01	0.05
	Fe K	99.99	99.95
	total	100	100

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
