# Peer review of "Fabrication and Tribological Properties of Mesocarbon Microbead–Cu Friction Composites"

_materials, 2020, doi:10.3390/ma13020463_

Round 1

Reviewer 1 Report

The article Materials-637944 "Fabrication and tribological properties of mesocarbon microbead-Cu friction composites" by Hai-Xia Guo and Jian-Feng Yang represents decent scientific level. In my opinion it is an interesting work concerning the tribological behaviour of copper based composites with addition of graphite flakes (GrF) and mesocarbon microbeads (MCMB) serving as solid lubricants. The emphasis of the research was placed on enhanced creation of cementite phase in presence of MCMB and its influence on mechanical properties of produced composites. Although, I found some major issues, that do not let me accept it for publication in the current shape. All of the aspects requiring explanations or corrections are listed below.

The entire manuscript has some major grammar and style mistakes. In addition, several sentences are way too long and many phrases are repeated in consecutive sentences, making them hard to read and understand. English of the whole paper will require significant revision. Authors should take the help of some reliable professional agency to proofread their manuscript or of a native English speaker with some technical knowledge of the subject. The appropriate changes should be done.

Authors are asked to write in colour any technical and language improvement made in their revised text.

Additionally, some essential issues that should be addressed are listed below.

The output powders used in the presented work lack some basic information: purity, producer designation, morphology. Why both the aluminium and alumina powders used in this study had such a different mean particle size compared to copper, iron and GrF/MCMB? Based on presented SEM images (Fig.1 a and c) and particle size of alumina, it is impossible to state if it is uniformly distributed within the matrix. Used magnification does not allow to conclude on possible voids/microcracks presence too. In addition, images in BSEM mode (Fig.1 b and d) are impossible to read. Higher quality SEM images should be attached with visible markers pointing out different phases. There are no comments for Si peaks on XRD pattern. There is also no diffraction analysis made from composite to prove the presence of additional phases. Authors conclude that there is formation of uniform and compact tribofilm, although there is no evidence of it in the study, except for reduced wear rate which is an outcome of cementite presence. There is lack of consistency in the described results, Authors use individual analysis to confirm their thesis.

Author Response

January 2, 2020

Dear reviewer,

Thank you very much for the reviewer’s comments about our manuscript entitled “Fabrication and tribological properties of mesocarbon microbead-Cu friction composites” (Manuscript ID: materials-637944). We have carefully studied the reviewer’s comments and accepted all of the suggestions presented in the review comments. The manuscript was improved accordingly, and all the revised and added parts were high-lighted by red color in the revised version.

Yours Sincerely,

Jianfeng Yang

Reviewer 2 Report

96: What was the cooling rate? And at what temperature did you remove the samples from the furnace? The cooling rate has a strong effect on microstructure of Fe-C alloys, and the differences during cooling of your samples may be the reason of hardness differences.

116: Correct the value of holding time.

142: This is Eq. (2).

159-175: I agree that during sintering at the temperature 980 C the carbon atoms may diffuse into the matrix and react with iron phase to form the pearlite. But the maximum possible quantity of Fe3C should be the same despite on the volume fraction of  GrF or MCMB. Maximum solubility of carbon in austenite at the temperature 980 C is about 1.7 wt.%. It gives about 25% of Fe3C. All non-reacted carbon atoms should be presented in the form of residual graphite. The structures on Fig.1 (b) & (d) look similar. To prove your opinion about influence of different reactivity of GrF and MCMB on properties of the sintered samples you must present additional results. Maybe, you have to perform the XRD analysis of sintered samples and the microstructure analysis with building of chemical distribution maps. 

213: Could you explain the presence of Si in GrF and MCMB?

You have an aluminum in your composite. What about the behavior of aluminum during sintering and it's influence on microstructure and properties of the composite materials?

Author Response

(The authors gave the same response as above.)

Round 2

Reviewer 1 Report

The article Materials-637944 "Fabrication and tribological properties of mesocarbon microbead-Cu friction composites" by Hai-Xia Guo and Jian-Feng Yang represents decent scientific level. In my opinion it is an interesting work concerning the tribological behaviour of copper based composites with addition of graphite flakes (GrF) and mesocarbon microbeads (MCMB) serving as solid lubricants. The emphasis of the research was placed on enhanced creation of cementite phase in presence of MCMB and its influence on mechanical properties of produced composites.

All of the previously listed issues were addressed by Author and either corrected, either clarified in revised manuscript.

For that reason I strongly recommend this manuscript for publication in Materials MDPI.

Author Response

January 14, 2020

Dear reviewer,

Thank you so much for reviewer's great efforts on reviewing our work and recommending our manuscript entitled “Fabrication and tribological properties of mesocarbon microbead-Cu friction composites” (Manuscript ID: materials-637944).

Yours Sincerely,

Jianfeng Yang

Tel.: +86-029-82667942-803;

E-mail: [email protected] (Jianfeng Yang);

Address: State Key Laboratory for Mechanical Behavior of Materials, Xi'an Jiaotong University, No. 28, Xianning West Road, Xi'an, Shaanxi, 710049, People’s Republic of China

Reviewer 2 Report

I want to thank the authors for the changes made. The paper became better. I have a minor remark to consider dividing the fig.2 into several parts, at least two.

Author Response

January 14, 2020

Dear reviewer,

Thank you very much for the reviewer’s comments about our manuscript entitled “Fabrication and tribological properties of mesocarbon microbead-Cu friction composites” (Manuscript ID: materials-637944). We have carefully studied the reviewer’s comments and accepted the suggestions presented in the review comments. The manuscript was improved accordingly, and the revised and added parts were high-lighted by red color in the revised version.

Yours Sincerely,

Jianfeng Yang

Tel.: +86-029-82667942-803;

E-mail: [email protected] (Jianfeng Yang);

Address: State Key Laboratory for Mechanical Behavior of Materials, Xi'an Jiaotong University, No. 28, Xianning West Road, Xi'an, Shaanxi, 710049, People’s Republic of China
